# First Detection of NADC34-like PRRSV as a Main Epidemic Strain on a Large Farm in China

**DOI:** 10.3390/pathogens11010032

**Published:** 2021-12-29

**Authors:** Chao Li, Bangjun Gong, Qi Sun, Hu Xu, Jing Zhao, Lirun Xiang, Yan-Dong Tang, Chaoliang Leng, Wansheng Li, Zhenyang Guo, Jun Fu, Jinmei Peng, Qian Wang, Guohui Zhou, Ying Yu, Fandan Meng, Tongqing An, Xuehui Cai, Zhi-Jun Tian, Hongliang Zhang

**Affiliations:** 1State Key Laboratory of Veterinary Biotechnology, Harbin Veterinary Research Institute, Chinese Academy of Agricultural Sciences, Harbin 150001, China; lichao2459@foxmail.com (C.L.); bangjungong@foxmail.com (B.G.); S77arch@163.com (Q.S.); xuhu1995@foxmail.com (H.X.); zhaojing94vet@163.com (J.Z.); zhhlxlr@163.com (L.X.); tangyandong@caas.cn (Y.-D.T.); wansheng1994@163.com (W.L.); zhenyang499@163.com (Z.G.); fujunl@foxmail.com (J.F.); pjm7614@163.com (J.P.); wangqian@caas.cn (Q.W.); zhouguohui@caas.cn (G.Z.); mengfandan@caas.cn (F.M.); antongqing@caas.cn (T.A.); caixuehui139@163.com (X.C.); tianzhijun@caas.cn (Z.-J.T.); 2Henan Key Laboratory of Insect Biology in Funiu Mountain, Henan Provincial Engineering Laboratory of Insects Bio-Reactor, China-UK-NYNU-RRes Joint Laboratory of Insect Biology, Nanyang Normal University, Nanyang 473061, China; lenghan1223@126.com; 3Department of Preventive Veterinary Medicine, College of Veterinary Medicine, Qingdao Agricultural University, Qingdao 266109, China; yuyingeng@163.com

**Keywords:** NADC34-like PRRSV, main epidemic strains, 100 aa deletion, RFLP

## Abstract

The newly emerged sublineage 1.5 (NADC34-like) porcine reproductive and respiratory syndrome virus (PRRSV) has posed a direct threat to the Chinese pig industry since 2018. However, the prevalence and impact of NADC34-like PRRSV on Chinese pig farms is unclear. In the present study, we continuously monitored pathogens—including PRRSV, African swine fever virus (ASFV), classical swine fever virus (CSFV), pseudorabies virus (PRV), and porcine circovirus 2 (PCV2)—on a fattening pig farm with strict biosecurity practices located in Heilongjiang Province, China, from 2020 to 2021. The results showed that multiple types of PRRSV coexisted on a single pig farm. NADC30-like and NADC34-like PRRSVs were the predominant strains on this pig farm. Importantly, NADC34-like PRRSV—detected during the period of peak mortality—was one of the predominant strains on this pig farm. Sequence alignment suggested that these strains shared the same 100 aa deletion in the NSP2 protein as IA/2014/NADC34 isolated from the United States (U.S.) in 2014. Phylogenetic analysis based on open reading frame 5 (ORF5) showed that the genetic diversity of NADC34-like PRRSV on this farm was relatively singular, but it had a relatively high rate of evolution. Restriction fragment length polymorphism (RFLP) pattern analysis showed that almost all ORF5 RFLPs were 1-7-4, with one 1-4-4. In addition, two complete genomes of NADC34-like PRRSVs were sequenced. Recombination analysis and sequence alignment demonstrated that both viruses, with 98.9% nucleotide similarity, were non-recombinant viruses. This study reports the prevalence and characteristics of NADC34-like PRRSVs on a large-scale breeding farm in northern China for the first time. These results will help to reveal the impact of NADC34-like PRRSVs on Chinese pig farms, and provide a reference for the detection and further prevention and control of NADC34-like PRRSVs.

## 1. Introduction

Since 2018, China, as the world’s largest pig producer, has not only faced the threat of classical swine fever virus (CSFV), porcine circovirus 2 (PCV2), pseudorabies virus (PRV), and porcine reproductive and respiratory syndrome virus (PRRSV), but has also suffered a heavy blow from African swine fever virus (ASFV) [1,2,3]. PRRSV is recognized as an important production disease worldwide, and is clinically characterized by reproductive failure in sows, including abortion and elevated fetal losses, as well as respiratory disorders in pigs of all ages, leading to elevated mortality and poor growth performance—especially in weaning and nursery herds [1,4]. Because current PRRSV vaccines display varying protection against homologous and heterologous challenges, the diversity of wild-type PRRSV variants makes it difficult to predict the nature of immunity elicited by naturally occurring variants against heterologous challenges [5,6,7], causing great economic losses to the global pig industry.

PRRSV belongs to the genus *Betaarterivirus*, family *Arteriviridae*, and order *Nidovirales* [8,9]. PRRSV is currently classified into two distinct species: *Betaarterivirus suid 1* (PRRSV-1) and *Betaarterivirus suid 2* (PRRSV-2) [9]. Both PRRSVs have been circulating in China for decades, and have caused substantial economic losses in the Chinese pig industry. PRRSV-2 has been the main epidemic strain in China, and is further classified into nine lineages based on open reading frame 5 (ORF5) sequences [10]. Four lineages (lineages 1, 3, 5, and 8) of PRRSV-2 circulate in the field [11]. In recent years, the dominant lineages of PRRSV-2 have shifted from sublineage 8.7 (CH-1a-like and HP-PRRSV-like) to sublineage 1.8 (NADC30-like).

In 2014, the ORF5 RFLP 1-7-4 lineage was prevalent in the United States (U.S.), and was recognized to be the cause of dramatic abortion “storms” in sow herds and high mortality among piglets [12,13]. Subsequently, that sublineage was confirmed to be endemic in Peru [14]. In China, NADC34-like PRRSV was first detected in Liaoning Province in 2017 [15]. Subsequently, the presence of such strains was also detected in southern China [16]. In 2020 and 2021, the detection of sublineage 1.5 (NADC34-like) and research on its spread were reported [17,18,19]. Unlike American NADC34-like PRRSVs, Chinese NADC34-like PRRSVs are currently mildly or moderately pathogenic in pigs [20,21]. However, the prevalence and impact of NADC34-like PRRSVs on Chinese pig farms is unknown. In this study, PRRSV was monitored for one production cycle (150 days per cycle) on a fattening pig farm. The breeding scale of this farm is 3400 heads. We studied the epidemic process and molecular characteristics of NADC34-like PRRSV from this pig farm in detail.

## 2. Results and Discussion

### 2.1. Viral Detection

In total, 412/3400 pigs died over the course of 150 days, and the death rate was 12.12% (Appendix A). A total of 46 piglets died in the first 15 days, artificially defined as stage 1 (Figure 1). The main clinical symptoms were loss of appetite and fever, followed by acute death, and the main pathological changes observed during necropsy of piglets were intestinal hemorrhage, abdominal hemorrhage, and peritoneal effusion. These deaths were preliminarily judged to be caused by bacterial infection based on clinical symptoms, the subsequent test results, and the effect of antibiotics (Appendix A). On the 10th day, antibiotics were given urgently (drug sensitivity test showed that cefoperazone–sulbactam was effective) for prevention and treatment, and the symptoms were significantly relieved. The peak mortality period (287/412) occurred from the 16th day to the 45th day, artificially defined as stage 2 (Figure 1). Most pigs showed obvious clinical respiratory symptoms, such as cough, wheezing, or diaphragmatic breathing. Antibiotics (cefoperazone–sulbactam, soluble powder of doxycycline hydrochloride, tilmicosin solution, and neomycin sulfate) were used again for prevention and treatment, but the effect was not satisfactory. Necropsy of the dead pigs showed lung consolidation, partial intestinal bleeding, and abdominal hemorrhage. We assumed that these symptoms were caused by a viral infection based on clinical symptoms and the subsequent test results. The number of dead pigs (79/412) decreased after the 45th day, artificially defined as stage 3 (Figure 1a), and the majority of the deaths occurred in pigs previously isolated for respiratory symptoms. To explore the causes of death in the piglets or pigs, a total of 283 samples were collected from dead pigs and tested for ASFV, CSFV, PRRSV, PRV, and PCV2. PRRSV and PCV2 were detected, while ASFV, CSFV, and PRV were not (Appendix A). During the three different stages noted above, the detection rates of PRRSV and PCV2 were 17.39% (8/46), 33.13% (53/160), and 29.87% (23/77); and 89.13% (41/46), 81.88% (131/160), and 80.52% (62/77), respectively (Figure 1a). Based on the above results, we speculate that death in the early stages may have been mainly due to bacterial infection, and that PRRSV contributed to the death curve on this pig farm, but PCV2 did not.

### 2.2. Phylogenetic Analysis

To explore the relationship between the PRRSV subtype and the death of pigs on the farm, we sequenced the NSP2 and ORF5 genes for all PRRSV-positive samples via the Sanger method. A total of 78 NSP2 sequences and 69 ORF5 sequences were obtained; of these, 54 samples were NADC30-like PRRSV (64.29%), 17 were NADC34-like PRRSV (20.24%), 11 were HP-like PRRSV (13.10%), 1 was CH-1a-like PRRSV (1.19%), and 1 was QYYZ-like PRRSV (1.19%) (Figure 1b). Therefore, NADC30-like PRRSV, NADC34-like PRRSV, and HP-like PRRSV were the main epidemic strains on this farm. Furthermore, the detection rates of the main epidemic strains were 12.50%, 71.70%, and 65.22% (NADC30-like PRRSV); 0.00%, 26.42%, and 13.04% (NADC34-like PRRSV); and 87.50%, 1.89%, and 17.39% (HP-like-PRRSV) in the three different stages, respectively. Surprisingly, the outbreak times of NADC30-like and NADC34-like PRRSVs were consistent with the peak periods of pig deaths on the pig farm (Figure 1). The above results demonstrate that NADC30-like PRRSV and NADC34-like PRRSV—but not HP-like PRRSV—were closely related to the deaths of pigs on this farm. In addition, a number of NADC34-like strains (14/17) were detected in the subsequent 15 days after initial detection in stage 2. The spread of NADC34-like PRRSV seemed to be faster than that of NADC30-like PRRSV. The virulence levels previously reported in China were mild and moderate, and the virulence levels of NADC34-like PRRSV previously reported in other countries were uneven [13,20,21]. The pathogenicity of NADC34-like PRRSV on this farm remains to be studied.

### 2.3. Sequence Analysis of NADC34-like PRRSVs

The emergence of NADC34-like PRRSV was first reported in Liaoning Province, China, in 2017 [15]. This strain subsequently emerged in other provinces of China [16,17,18,19,21]. The positive samples were sequenced via the Sanger method. All of the NSP2 sequences of the NADC34-like strains shared the same 100 consecutive amino acid deletions between 328 and 427 as previously reported, compared with ATCC-VR2332 (accession number: U87392) (Figure 2). These deletions can be used as molecular markers to distinguish NADC34-like strains from other type 2 PRRSV strains in China, similar to the consistent NSP2 protein deletion pattern in NADC30-like PRRSV [22]. The amino acid identities of NSP2 of the NADC34-like strains on this farm were between 99.2% and 99.9%. In the NCBI library, the highest identity was with IA/2014/NADC34 (accession number: MF326985), at 92.6%.

The nucleotide identity of the NADC34-like PRRSV ORF5 gene on this pig farm was 99.2–100%, and also had the highest identity with the IA/2014/NADC34 strain in the NCBI database, along with a nucleotide similarity of 96.9–97.2%. The consistency between these strains and the first NADC34-like strain reported in China (LNWK130) was 94.9–95.0%. These results indicate that NADC34-like PRRSV has evolved in China. Combined with the NSP2 analysis of NADC34-like PRRSV, NADC34-like PRRSV infection on this pig farm was caused by a single strain; this provides a good platform for studying the evolution rate of the NADC34-like strain [23]. Many methodological approaches previously used to study PRRSV ignored the fact that the evolutionary and epidemiological dynamics of rapidly evolving pathogens—such as PRRSV—occur on approximately the same timescale. Thus, they must be studied under a unified methodological setting in order to be properly understood and to prevent biased conclusions, subsequently improving related decision-making processes [12,24]. The NADC34-like strains on this farm showed a strong time signal (the correlation between the genetic difference and sampling time r^2^ was 0.52), and were thus suitable for phylogenetic analysis involving a molecular clock. The estimated viral substitution rates were 3.1 × 10^−2^ substitutions/site/year—higher than the evolution rate, which ranged from 6.6 × 10^−3^ to 1.3 × 10^−2^ substitutions/site/year for all subtypes of lineage 1 previously reported in the U.S. [7,12]. This is an alarming finding that indicates that the time from the appearance of NADC34-like PRRSV in China to its peak in the population was shorter than that in the U.S. (4.5 years on average) [7]. Moreover, surveillance of PRRSV showed that the number of NADC34-like PRRSVs has obviously increased since 2020—especially in 2021 (unpublished data). Therefore, we speculate that NADC34-like PRRSV has become dominant in parts of China.

We further classified the NADC34-like PRRSV strains on this farm according to restriction fragment length polymorphism (RFLP) of the ORF5 gene [25,26,27]. The RFLP pattern of ORF5 of TZJ1277 is 1-4-4, while those of the others are 1-7-4. Compared with the newly emerged PRRSV lineage 1C variant (MW887655) in the U.S., TZJ1277 has a closer relationship with the previously reported NADC34-like PRRSV (Figure 3a). RFLP typing has recognized shortcomings, which include an inability to represent genetic relationships between different RFLP types, the potential for distantly related viruses to share the same RFLP type, and instability of RFLP types over as few as 10 animal passages [25,28]. Partially due to these ambiguities in the interpretation of RFLP types, researchers in many countries have formulated their own naming conventions based on the epidemic situation in their countries [7,28], so the classification of viral strains in their home countries is crucial.

### 2.4. Whole Genome Analysis

Whole-genome sequencing (WGS) can reveal more information about PRRSV than traditional Sanger sequencing analysis of ORF5 [29]. To explore the evolutionary relationships of all of the isolated PRRSV strains with the representative strains, we sequenced two complete genomes from NADC34-like PRRSVs from this farm. According to our earlier research, NADC34-like PRRSV strains in China can be divided into two groups (A and B) based on genome-wide phylogenetic analysis [17]. WGS phylogenetic tree analysis showed that two full-length sequences measured (TZJ864 and TZJ921) were clustered into group A (Figure 3b). All group A sequences were from Heilongjiang Province, China [17,18], which is one of the regions where NADC34-like strains appeared earlier.

To date, six commercial PRRSV vaccines—CH-1R, JXA1P80 HuN4-F112, GDr180, TJM-F92, and RespPRRS MLV, based on their corresponding strains—have been widely used in China [30,31]. These live, attenuated vaccines are all derived from lineage 5 or lineage 8, and are distantly related to NADC34-like strains (sublineage 1.5). Pigs are immunized with CH-1R MLV vaccine one week after entering the farm. The nucleotide homology of TZJ864 and TZJ921 with CH-1R was only 83.3%. Additionally, it has been reported that the existing attenuated PRRSV vaccine does not provide good protection effects against NADC30-like PRRSV (sublineage 1.8) strains that belong to lineage 1, such as NADC34-like PRRSV (sublineage 1.5) [30,32]. Therefore, the development and application of lineage-1-virus-related vaccines are urgently needed.

### 2.5. Recombination Analysis

Recombination is an important evolutionary strategy for PRRSV. Both interlineage and intralineage recombination occurred, contributing to the emergence of new epidemic isolates [11]. The American 1-4-4 lineage1C PRRSV variant is based on IA14737-2016 (NADC34-like) as the parent strain, with IA/2014/NADC34 and NADC30 strains providing recombinant fragments. PRRSV recombination events have been the focus of researchers [13,33,34]. Recombinant PRRSVs have been increasingly isolated since NADC30-like PRRSVs emerged in China [14,35,36]. To explore whether the NADC34-like PRRSV isolated from this farm was a recombinant virus, we performed a recombination analysis on the two NADC34-like PRRSVs (TZJ864 and TZJ921) obtained from the farm. Recombination analysis and sequence alignment showed that they were not recombinant viruses, and had relatively high identity (98.9%). However, the coexistence of multiple PRRSV lineages in swine herds provides suitable conditions for recombination of PRRSV in the field. Coincidentally, we have detected many recombinant NADC34-like PRRSVs from other pig farms since June 2021 (data unpublished). The characteristics and virulence of these recombinant NADC34-like PRRSVs need to be further studied.

Concerningly, the U.S. has reported high economic losses due to NADC34-like reorganization [13]. Importantly, NADC34-like and NADC30-like PRRSVs recombine with strains of different subtypes, resulting in inconsistent virulence among the recombinant strains, and causing great obstacles in the prevention of PRRSV [36,37]. Considering that NADC30-like PRRSV has become the main epidemic strain in China [38], outbreaks of NADC34-like PRRSVs on pig farms will inevitably lead to more frequent fragment exchanges between them. Therefore, we need to increase awareness of the importance of continuous monitoring of NADC34-like PRRSV strains, strictly control the selection of breeding pigs, and prevent the occurrence of multiple subtypes of PRRSV on pig farms.

## 3. Materials and Methods

### 3.1. Farm Information

A finishing pig farm (3400 heads) where five pathogens (ASFV, CSFV, PRV, PRRSV, and PCV2) had been monitored was selected for the study. The farm is located in Heilongjiang Province, China, and there are no neighboring pig farms within 3 km. The farm has its own boar stud located 5 km away from the breeding herd, which is immunized with CH-1R MLV. Blood is collected from piglets before their transport to the farm to detect PRRSV by RT-PCR. The 3400 piglets (6–7 weeks old) enter the farm at same time. No testing was performed after arriving at the pig farm until death. The finishing pig farm adopts an all-in, all-out fully enclosed management model; personnel and materials entering and leaving the area are disinfected, and the manure used to produce fertilizer is treated appropriately. The internal layout of the pig farm is reasonable—the buildings are at least 6 m apart, ventilation is good, disinfection is performed regularly, and necessary vaccines are administered in a timely manner. Livestock are vaccinated with the classical PRRSV modified live virus (MLV) vaccine (CH-1R strain), classical swine fever MLV vaccine (C strain), pseudorabies virus MLV vaccine (Bartha k61 strain), foot and mouth disease (FMD) bivalent inactivated vaccine (Re-O/MYA98/JSCZ/2013 Zhu + Re-A/WH/09 Zhu), and porcine circovirus 2 inactivated vaccine (LG). The pig farm employs professional management and veterinary personnel.

### 3.2. Sample Collection and Data Processing

Approximately 15–45 lung and submandibular lymph node samples from livestock on the farm were submitted for laboratory testing approximately every 15 days during the study period (Appendix A). During the breeding period, dead pigs were dissected to collect samples, and the time of death and number of deaths were recorded. The number of dead pigs was counted every day and summed every 15 days. From September 2020 to January 2021, 283 clinical samples were collected from 412 dead pigs on the farm. Tissue sample disposal, RNA and DNA extraction, cDNA preparation, RT-PCR analysis, and genome sequencing were conducted as previously described [39,40,41,42,43]. The primers used for viral detection (ASFV, CSFV, PCV2, PRRSV, and PRV) and complete genome amplification have been reported previously [40,41,44,45,46,47]. The positive samples were sequenced via the Sanger method.

### 3.3. Phylogenetic Analysis

All reference strains were downloaded from the NCBI database, and the corresponding sequences were compared and interpreted. Deduced amino acid sequences were aligned with ClustalW with Lasergene software. All sequences were aligned using MAFFT version 7 [48], with default parameters, and manually adjusted in MEGA6 [49]. We followed the same rationale for the classification of sequences into lineages as previously published [10]. To identify evolutionary relationships between strains on this farm, phylogenetic trees were constructed in MEGA 6.0 by the neighbor-joining method with a bootstrap value of 1000 replicates, and with the Kimura two-parameter substitution model. The trees were annotated and modified using Evolview (version 2.0) (https://www.evolgenius.info/evolview/#login accessed on 12 November 2021) [50]. To analyze recombination events, similarity analysis was performed using SimPlot software (v3.5), with a 200 bp window and a 20 bp step [33].

### 3.4. Estimation of the Evolutionary Rate

The temporal signal in the phylogenetic datasets of NADC34-like PRRSV was first investigated using TempEst to confirm the appropriateness of the data for time-scaled phylogenetic tree reconstruction [51]. To estimate the evolution rate of the NADC34-like PRRSV strain on this pig farm, a relaxed uncorrelated lognormal (UCLN) molecular clock was used, with a flexible Bayesian Skygrid plot (BSP) demographic model and a general-time reversible model of nucleotide substitution with gamma-distributed rate variation among sites (GTR+γ, allowing for partitions into codons in any of the three positions [23]. The Markov chain Monte Carlo (MCMC) algorithm was run for 200 million steps and sampled every 20,000 steps. Convergence was assessed with effective sample size (ESS) values, and ESS values over 200 were considered adequate. These analyses were performed using BEAST (v1.10.4). Three independent runs were performed in this study to prevent any local convergence. The BEAST results were entered into Tracer to evaluate model convergence and consistency between replicates.

### 3.5. Statistical Analysis

GraphPad Prism 8.0 (San Diego, CA, USA) was used to perform the statistical data analyses.

## 4. Conclusions

In summary, multiple subtypes of PRRSV coexist on this pig farm. NADC34-like PRRSV, which was correlated with the death curve, was first reported to be one of the main endemic strains on a pig farm in Heilongjiang Province, China. NADC34-like PRRSV from this farm had high nucleotide similarity and a 100 aa deletion in the NSP2 protein, but no recombination. The genetic diversity of NADC34-like PRRSV is relatively singular, but it has a relatively high evolution rate. The prevalence of NADC34-like PRRSV on other pig farms in China needs further attention.

## Figures and Tables

**Figure 1 pathogens-11-00032-f001:**
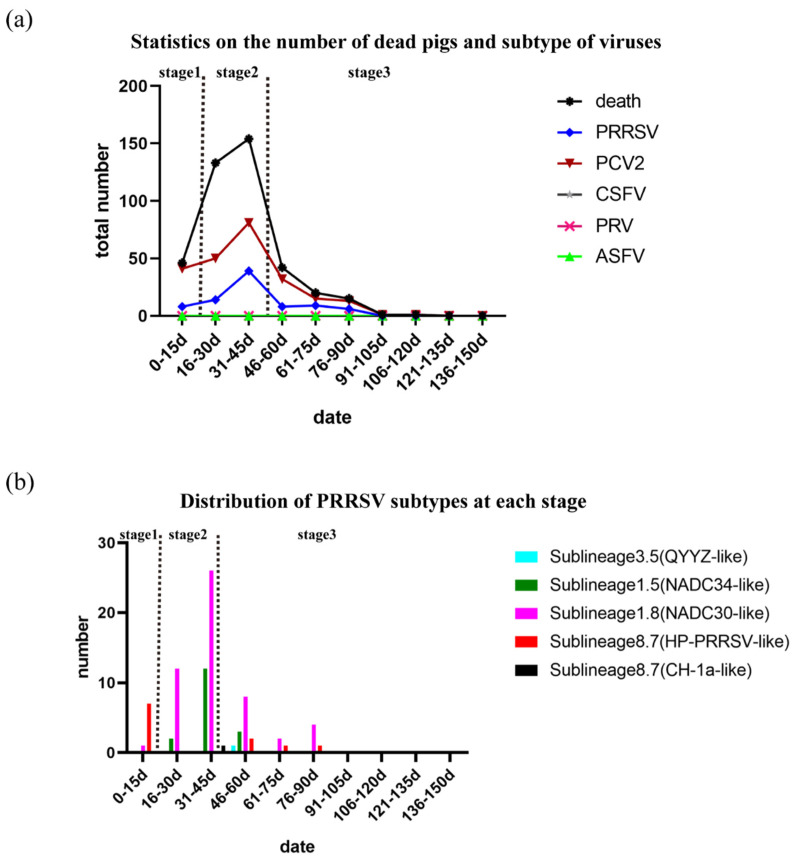
Statistics on the number of dead pigs and subtypes of viruses: (**a**) Statistics on the numbers of dead pigs and viruses detected. The number of dead pigs and the number of viruses detected are represented by different colors. (**b**) Distribution of PRRSV subtypes at each stage. Sublineage 1.5 is represented by dark green; sublineage 1.8 is represented by bright purple; sublineage 3.5 is represented by sky blue; and sublineage 8.7 is represented by red (HP-PRRSV-like) and black (CH-1a-like).

**Figure 2 pathogens-11-00032-f002:**
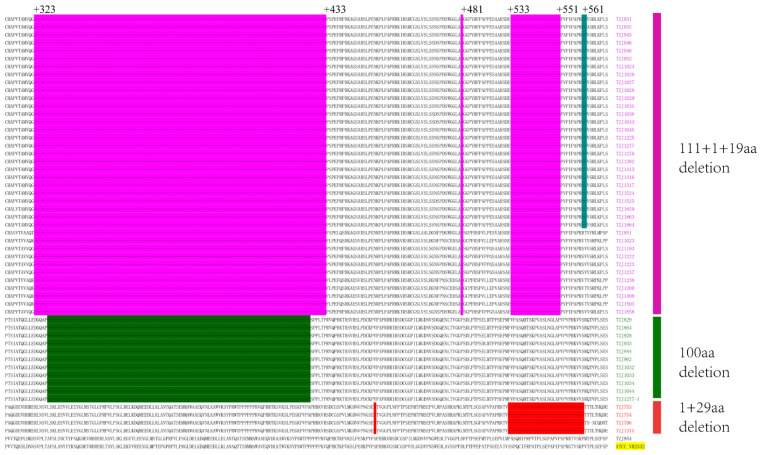
Alignment of the deduced amino acid sequence based on the NSP2 gene. ATCC-VR2332 is the reference. NADC30-like PRRSVs were identified as a 111 aa deletion at positions 323–433, a 1 aa deletion at position 481, and a 19 aa deletion at positions 533–551. All of the NADC34-like PRRSVs had 100 aa deletions corresponding to positions 328–427 (dark green regions). All of the HP-like PRRSVs had a 1 aa deletion at 482 and 29 aa deletions at 533–561.

**Figure 3 pathogens-11-00032-f003:**
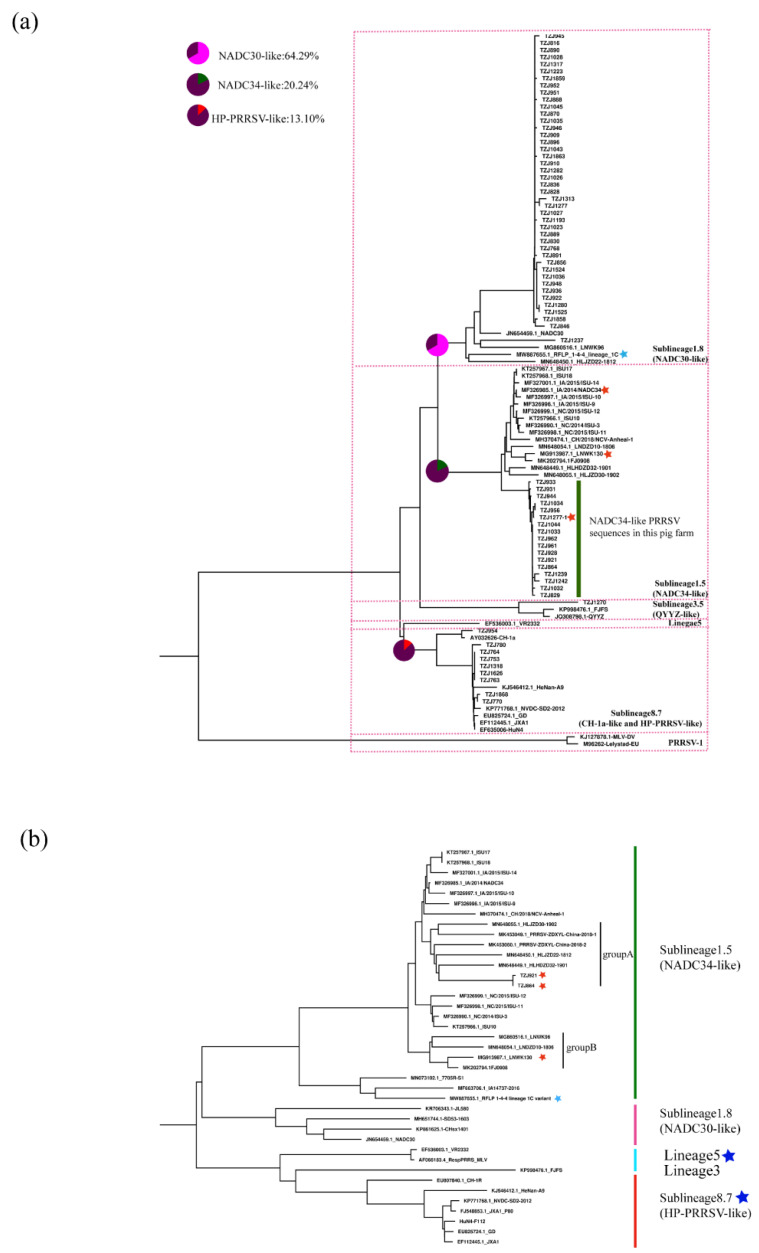
Phylogenetic analysis of NADC34-like PRRSV: (**a**) Phylogenetic analysis of PRRSVs based on the ORF5 gene. The percentage of positives among different strains is represented by a pie chart. (**b**) Phylogenetic tree of PRRSVs based on full-length genomes. The American 1-4-4 PRRSV strains are labeled with blue stars; corresponding types of PRRSV vaccine currently on sale in China are labeled with dark blue stars.

## Data Availability

All data pertaining to the study described in the manuscript are described in the report.

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
