# Peer review of "First Detection of NADC34-like PRRSV as a Main Epidemic Strain on a Large Farm in China"

_pathogens, 2021, doi:10.3390/pathogens11010032_

Round 1

Reviewer 1 Report

The abstract introduces the study as following seven pig farms but there are only results for one farm.  Please either change your abstract or include the results from the other farms.  Data from one farm does not provide any information on prevalence.

INTRODUCTION

You need to introduce the viruses that you look for in the pigs - why are they important to the pig industry, how widespread are they worldwide/in China, what is the mortality rate in adult pigs and piglets?  Then you need to focus on PRRSV and describe what is known about the clinical signs, disease progression and pathology. 

You mention in your methods that the pigs are vaccinated against PRRSV - does this vaccine not provide protection against all strains?

What is the background of the farm in this study?  you mention in the methods that this farm had previously had five swine viruses detected there - how long ago?  where do the pigs come from?  Are they screened when they arrive?

RESULTS

Line 65 - Please use "pig/piglet" accurately - pig for adults only and "piglets" for young pigs only.

Line 65 - Please indicate what percentage 412 pigs represents of the entire pig population.  Also what percentage of the entire piglet population.

Line 72 - you are describing pathology and clinical symptoms for the second time in the same paragraph. are you trying to show that symptoms and pathology are different in pigs that died early compared to those at later timepoints?

Figure 1 - this is a really hard graph to understand.  please show the data differently, perhaps using two separate graphs. what are the definitions of the "stages"?  the pie graphs need to be included in the figure legend.

Line 77 - what kind of samples were collected?  how many pigs were sampled? what method was used to detect the viruses?  were any samples co-infected?  why was diagnostics for a bacterial infection not done?  all this information should be included in a table and made transparent. 

Line 81-83 - how did you come to this conclusion? i don't see any data or explanation that supports this statement.

Figure 2 - the legend mentions a blue region but there is no blue region. Please describe where the sequences came from. Is each sequence from a different pig? What kind of sample did they come from?  did you find sequences from any other viruses that were not initially included in the diagnostic panel?

DISCUSSION

How do you findings compare to other studies on NADC34 PRRSV?   is the pathology similar?  are the death rates similar?

METHODS

Line 205 - which lymph node?

CONCLUSION

Line 246 - you have not shown in this paper that multiple types of PRRSV commonly exist on pig farms. you have only studied a single farm in this paper.

I would argue that NADC30 and NADC34 are both equally common at peak mortality times.  What about the high prevalence of HP-PRRSV in stage 1 of your study?

Reviewer 2 Report

I reviewed the manuscript entitled “First detection of NADC34-like PRRSV as a main epidemic 2 strain on a large farm in China”. In this study authors describe the prevalence and relevance of NADC34-like PRRSV in regions of China.

Overall, I consider that the methodology employed in this study supports the results and conclusions stated in this communication.

Some suggestions to improve this communication would be to include some evolutionary analysis to report potential sites evolving under positive selection in the epidemic lineage. It may be relevant considering the lack of recombination reported in this lineage. How different are the two sequences associated with the epidemic lineage in comparison with other sequences reported in China, including the sequences of the vaccines used in China? I would suggest to include this information in the results and speculate in the discussion.

Round 2

Reviewer 1 Report

Line 70 - could you please indicate why this study period of 150 days was chosen?  was this time period chosen because of the deaths occurring?  or was it set up prior to knowing about the deaths?

Line 70 - it would also be helpful to introduce the background of the pigs at this point.. please describe how long the pigs were on the farm, where they came from, and whether they were tested for any disease upon arrival so that the epidemiology is clear.  you answered this question previously, but this would be an ideal location for it to appear in the manuscript.

Line 70 - you've also previously published a paper about detecting PRRSV on 16 farms in China - why have you not referenced it?

Line 75 - thank you for your pig/piglet changes but it's still not completely clear to me how many of each are present.  Perhaps the authors could state how many total pigs and how many total piglets there were on the farm, and then state how many of each died during the study period.

Line 76 - the supplementary table is good however i am unclear why your  number of deaths are higher than the number of samples - are the titles reversed?  the table also needs a figure legend.  does sata mean "stage"?  perhaps this could be changed.

Line 81 - The data in supplementary table 2 is great, however, is incomplete.  could you please define the abbreviations used in the headings, and provide more detail about the samples.  I am wondering why there are ~280 samples when only 15 piglets died early, and there are 412 piglets in total (it just doesn't match).  perhaps you could add a column indicating when the day of death occurred for each sample and the age group of the pig or piglet (suckling, weaned, finishing, sow). i believe a figure legend for the table is missing as well - the abbreviations used in the column titles should be defined here and the study day when the testing was done.

Line 89, 92, 95 etc - pigs or piglets?  PRRSV is less lethal in adult pigs so this is important.

were the mothers of the dead piglets tested for virus infection or antibodies to the viruses?  can you speculate where the infection came from originally?

Figure 1 - thank you for modifying the figure, it is much better. looking at the data, i thought it might be interesting to also include a table indicating the same data, but separated based on age group.  for example, for pigs it is common to distinguish between suckling piglets, weaned piglets, finishing pigs and sows. sorry to be picky, but the red and orange colors are hard to distinguish - could this be changed?

Line 122 - is there experimental data on HP infections?  can it be lethal?  why is it so common in the group of pigs dying of bacterial infection? does this support your conclusion?

Line 127 - could you please provide some data and the range of results?  For example, Germany has reported the lowest virulence rate (33%) while in the Phillipines rates as high as 67% have been detected.

Line 129 - please mention here that you are using Sanger sequencing and therefore would detect pigs that have coinfections with more than one type of PRRSV.
